

# Aircraft Observations of Continental Pollution In the Equatorial Lower Stratosphere over the Tropical Western Pacific During Boreal Winter

Jasna V. Pittman[1], Bruce C. Daube[1], Steven C. Wofsy[1], Elliot L. Atlas[2], Maria A. Navarro[2*], Eric J. Hintsa[3,4], Fred L. Moore[3,4], Geoff S. Dutton[3,4], James W. Elkins[4], Troy D. Thornberry[4,5], Andrew W. Rollins[5], Eric J. Jensen[5], Thaopaul Bui[6], Jonathan Dean-Day[6], Leonhard Pfister[7]

[1]Harvard John A. Paulson School of Engineering and Applied Sciences, Harvard University, Cambridge, MA, 02138, U.S.A.
[2] Department of Atmospheric Sciences, Rosenstiel School of Marine, Atmospheric, and Earth Science, University of Miami, Miami, FL, 33149, USA
[3] Cooperative Institute for Research in Environmental Sciences, University of Colorado, Boulder, 80309, U.S.A.
[4] Global Monitoring Laboratory, NOAA, Boulder, CO 80305, U.S.A.
[5] Chemical Sciences Laboratory, NOAA, Boulder, CO 80305, U.S.A.
[6] Bay Area Environmental Research Institute, Moffett Field, CA, 94035, USA
[7] NASA Ames Research Center, Moffett Field, CA, 94035, USA
[*] Deceased

*Correspondence to*: Jasna V. Pittman (jasna@g.harvard.edu)

**Abstract.** Recent studies hypothesize that emissions from fires reaching the stratosphere can provide aerosols and aerosol precursors that initiate stratospheric ozone loss and lead to radiative heating of the stratosphere and cooling of the surface. Air from the troposphere enters the stratosphere primarily over the tropical western Pacific (TWP) during boreal winter. We report observations in the TWP of persistent, ubiquitous continental pollution in the tropical tropopause layer (TTL) and lower stratosphere (LS) during the Airborne Tropical TRopopause EXperiment (ATTREX) campaign in February-March 2014. We found concentrations of carbon monoxide (CO) enhanced up to 65 % over background levels in the deep tropics (5° S–15° N, 16 –17 km). Correlations of CO with hydrocarbon and halocarbon species indicated a biomass burning source, with the largest CO enhancements found in warmer, clear air. Satellite observations of CO did not detect the thin pollution layers observed by the aircraft, but did indicate Africa, Indonesia, and the western/central Pacific as geographical hot spots for CO in the TTL. Backward trajectories identified convective encounters in these areas as the dominant sources of polluted air in the TWP. Africa and Indonesia contributed about 60% of the excess CO, transported to the TWP in two to four weeks. Our study confirms that air in the TTL over the TWP is affected by emissions from distant fires that can rapidly reach the LS in the principal source region for air entering the stratosphere, supporting the view that fires in tropical regions could impact stratospheric ozone and temperatures.



## 1 Introduction

Emissions of long-lived ozone ($O_3$) depleting substances (ODS) of anthropogenic origin, such as chlorofluorocarbons (CFC) and hydrochlorofluorocarbons (HCFC), have contributed to significant thinning of the stratospheric $O_3$ layer via Chlorine- and Bromine-based chemistry (Solomon, 1999; Fahey et al., 2018, and references therein). Banning the production and emission

of ODS have helped reverse stratospheric $O_3$ loss with a current forecast for recovery to 1980 stratospheric $O_3$ values over the next four decades (Fahey et al., 2018; Amos et al., 2020).

In addition to the longer-lived CFCs and HCFCs, ODS include halogenated very-short-lived substances (VSLS). These compounds have atmospheric lifetimes of less than six months, which limit their buildup in the stratosphere. Fast transport via deep overshooting convection could provide a mechanism for rapid injection of these VSLS into the stratosphere,

enhancing the impact on stratospheric $O_3$ (Koenig et al., 2017; Oram et al., 2017; Wales et al., 2018; Filus et al., 2020; Tegtmeier et al., 2020; Treadaway et al., 2022). The impact of halogenated VSLS on the recovery of the stratospheric $O_3$ layer is uncertain, and a matter of concern because their emissions are unregulated and have continued to increase over time (Hossaini et al., 2017; Oram et al., 2017).

Recent studies have identified another mechanism for initiating $O_3$ loss that could potentially threaten the recovery

of the stratospheric $O_3$ layer (Bernath et al., 2022; Solomon et al., 2022 and 2023): increasing aerosols from wildfires. Aerosols induce stratospheric $O_3$ loss, mainly over polar latitudes, at very cold temperatures, involving condensed water vapor ($H_2O$), sulfate aerosols, nitric acid, halogens from ODS or VSLS, and solar ultraviolet (UV) radiation. Wildfire aerosols may contribute with not only sulfates, but also both primary and secondary organic aerosols that can take up hydrochloric acid, enabling chlorine activation to occur at warmer temperatures (Solomon et al., 2023). While still requiring halogens, addition

of wildfire aerosols to the stratosphere provides a new mechanism that could lead to more widespread stratospheric $O_3$ destruction no longer concentrated over polar latitudes during cold springtime conditions. As surface temperatures have risen, so have trends in frequency and intensity of fire weather throughout the world (Jones et al., 2022). Therefore, wildfire aerosols and aerosol precursors injected into the stratosphere could impede recovery of stratospheric $O_3$ as CFC concentrations decline.

Direct injection of smoke and biomass burning products into the lower stratosphere (LS) has been previously reported

in the extra-tropics (Jost et al., 2004; Cammas et al., 2009; Hooghiem et al., 2020; Peterson et al., 2021; Katich et al., 2023). At these higher latitudes, the descending motion of the Brewer-Dobson circulation dominates, limiting transport into the global stratosphere (Holton et al, 1995). Products of seasonal biomass burning have been frequently observed in the tropical upper troposphere (UT) as evidenced by elevated carbon monoxide (CO) (e.g., aircraft measurements - Ashfold et al., 2015, Anderson et al., 2016; Cussac et al., 2020; Lannuque et al., 2021; satellite observations - Schoeberl et al., 2006; Huang et al., 2012) and

a distinct mixture of trace gases as well as aerosols (Blake et al., 1996; Mauzerall et al., 1998; Andreae and Merlet, 2001; Gkatzelis et al., 2024). Although persistent features in the UT, these pollution events have not been reported in the higher-altitude tropical stratosphere, in part due to limited in situ sampling capabilities (Duncan et al., 2007; Rossow and Pearl, 2007).



Most air reaches the tropical stratosphere through the Tropical Tropopause Layer (TTL), a region ~4.5 km in depth, with a bottom at ~14 km (Fueglistaler et al, 2009). The TTL serves as the transition region from the convectively driven UT

to the radiatively dominated LS. Ascent rates throughout this transition layer are season dependent, with the fastest vertical transport into the tropical LS occurring during boreal winter (Fueglistaler et al, 2004; Park et al, 2010; Bergman et al., 2012). Reaching the high altitudes of the TTL requires strong convective lofting, which preferentially occurs over the longitudes of deepest convection, namely the Tropical Western Pacific (TWP), the Indian Ocean, Africa, and South America (Alcala and Dessler, 2002; Gettelman et al., 2002a; Fueglistaler et al., 2004; Bergman et al., 2012; Jensen et al., 2015). Within these

geographical regions, satellite observations show that the strongest and deepest convection occurs over the TWP during boreal winter (Gettelman et al, 2002a), making these longitudes a preferential entry point into the tropical stratosphere.

The present study examines the Airborne Tropical TRopopause EXperiment (ATTREX) dataset, which extensively profiled the TTL and the LS over the TWP during boreal winter. We investigate multiple trace gases of continental pollution, which we argue represent the influx of recent biomass burning products directly into the critical region for transport into the

global stratosphere, the TTL and LS over the TWP. We complement analysis of the aircraft dataset with satellite observations and backward trajectories to address three science questions. (i) Does continental pollution reach and measurably affect the composition of the TTL and LS over the TWP, the main gateway to the global stratosphere during boreal winter? (ii) What chemical trace gases are being brought by this pollution that could impact stratospheric aerosols and $O_3$? (iii) What are the characteristics of the transport processes involved (i.e., geographical origin, transport pathways, and transport timescales)?

This paper is structured as follows. Section 2 provides a description of the aircraft campaign, in situ measurements, satellite observations, and trajectory calculations. Section 3 characterizes the pollution plumes by exploring location within the TTL, frequency of observations, chemical composition, spatial and temporal context within the satellite record, geographic origin, transport pathways, and transit timescales to the TWP. Section 4 summarizes the conclusions of our study.

## 2 Methodology

### 2.1 Aircraft Campaign

The ATTREX campaign was one of the NASA Earth Venture Suborbital-1 investigations. This five-year project focused on studying physical and chemical processes in the TTL such as dehydration mechanisms, cloud formation, rates and pathways of vertical transport into the stratosphere, as well as the effects of tropical waves and deep convection on chemical composition (Jensen et al., 2017a). A payload consisting of three remote sensing and nine in situ instruments that measured chemical trace

gases, cloud properties, radiative fluxes, and meteorological conditions was installed on the remotely piloted NASA Global Hawk aircraft. No aerosol measurements were made during this campaign. The sampling strategy consisted of long-duration flights (18 to 24 hours each) continuously profiling the depth of the TTL (~14–19 km, 350–430 K, 150–70 hPa), reaching higher altitudes as the flight progressed. Measurements were made during two field deployments in the boreal winters of 2013 and 2014. The 2013 flights were based out of Edwards Air Force Base, CA (34.92° N, 117.89° W) and sampled the TTL over



the central and eastern Pacific in February-March 2013 (ATTREX-2). The 2014 flights were based out of Guam (13.39º N,

144.66º E) and sampled the TTL over the western Pacific in February-March 2014 (ATTREX-3). The full ATTREX dataset

provides over 280 flight hours and over 300 vertical profiles in the TTL covering a wide area of the Pacific Ocean: from 130º

E to 270º E in longitude, and across the Equator from 12º S to 35º N. This study focuses on a subset of the ATTREX dataset

in the deep tropics, between 12º S and 15º N over the western Pacific (ATTREX-3). Geographical coverage of all ATTREX

flights is shown in Fig. 1.

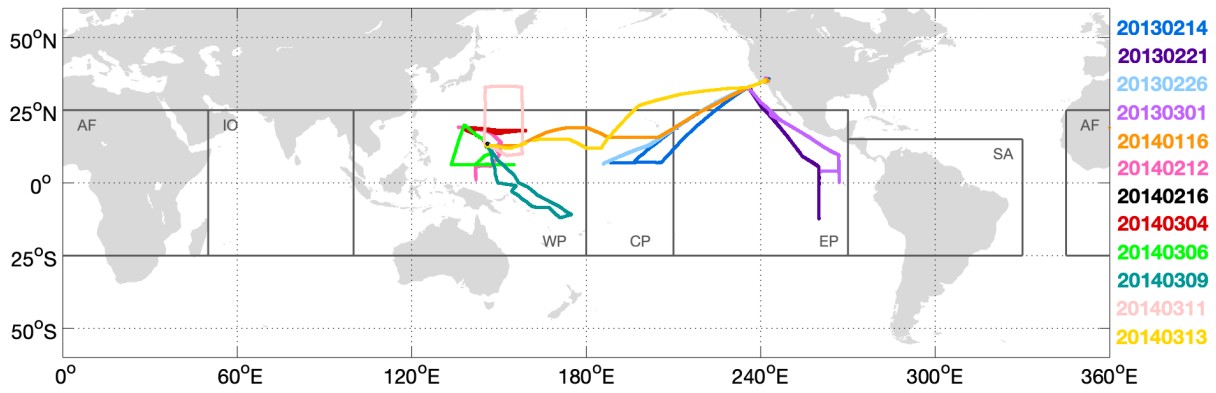

**Figure 1: NASA Global Hawk flights during ATTREX-2 (eastern and central Pacific, February–March 2013) and ATTREX-3 (western Pacific, February–March 2014). Definitions of geographical regions of convective origin based on backward trajectory calculations are as follows: AF (Africa, 0-50º E, 345-360º E), IO (Indian Ocean, 50-100º E), WP (Western Pacific, 100-180º E), CP**
**(Central Pacific, 180-210º E), EP (Eastern Pacific, 210-270º E), and SA (South America, 270-330º E).**

## 2.2 Aircraft Measurements

### 2.2.1 HUPCRS

Measurements of carbon dioxide ($CO_2$), methane ($CH_4$), and CO were obtained by the Harvard University Picarro Cavity

Ringdown Spectrometer (HUPCRS). The instrument was designed and built for autonomous operation on aircraft sampling in

the UT/LS region during the ATTREX campaign. It consists of a G2401-*m* Picarro gas analyzer (Picarro Inc., Santa Clara,

CA, USA) repackaged in a temperature-controlled pressure vessel, a separate calibration system with two multi-species gas

standards, and an external pump and pressure control assembly designed to allow operation at a wide range of altitudes. The

Picarro analyzer uses Wavelength-Scanned Cavity Ringdown Spectroscopy technology (Crosson, 2008; Rella et al., 2013;

Chen et al., 2013) to individually measure and report each trace gas within a ~2.2-second cycle.

The HUPCRS dataset consists of $CO_2$ and $CH_4$ for ATTREX-2 and $CO_2$, $CH_4$, and CO for ATTREX-3. Data are

reported every 10 seconds. Precision during flight was +/-0.2 parts per million by volume (ppmv) for $CO_2$ and +/-0.30 parts

per billion by volume (ppbv) for $CH_4$ during ATTREX-2, and +/-0.02 ppmv for $CO_2$, +/-0.20 ppbv for $CH_4$, and +/-2.5 ppbv



for CO during ATTREX-3. See the Supplement Section for a more detailed description of HUPCRS flight operation,
instrument schematics (Fig. S1), and in-flight performance (Figs. S2 and S3).

### 2.2.2 UCATS

Measurements of $O_3$ were obtained by the UAS (Unmanned Aircraft Systems) Chromatograph for Atmospheric Trace Species
(UCATS) package that contained a Model 205 UV photometer from 2B Technologies (Boulder, CO) (Hintsa et al., 2021). The
detection technique is based on direct absorption of 254 nm light following Beer-Lambert's Law. The instrument is a dual-
beam photometer that measures the ratio of light intensity between ambient air (no $O_3$ removed) and ambient air that is scrubbed
($O_3$ removed by manganese dioxide coated screens). Scrubbed and unscrubbed air alternatively pass through each cell as a
solenoid valve switches the flow paths every 2 seconds. Data are reported every 10 seconds with a precision of 5-10 ppbv. In
ATTREX-3, UCATS flew two Model 205 instruments, and reported the average of the two instruments to improve precision.

### 2.2.3 GWAS

A wide range of hydrocarbons and halogenated compounds were measured by the Global Hawk Whole Air Sampler (GWAS).
GWAS consists of a set of stainless-steel canisters filled with ambient air along the flight, with fill times ranging from 30
seconds at 14 km to 90 seconds at 19 km. Each canister is pressurized to 40-50 psia and analyzed in the laboratory, typically
within two to three weeks from collection time, using a multi-channel GC/MS/FID/ECD system (Agilent 7890 GC, 5973 MS)
(Andrews et al., 2016; Schauffler et al., 1999). A set of 90 canisters was used per flight. Due to aircraft power limitations,
GWAS pumps had to be turned off during descents, so sample collection took place during ascents only. GWAS reports
numerous compounds including chlorofluorocarbons, hydrochlorofluorocarbons, halocarbons, hydrocarbons, and halogenated
VSLSs. The present analysis focuses on GWAS trace gases with a wide range of atmospheric lifetimes (from a few days to
decades) and of source origin (e.g., combustion, industrial emissions, marine emissions, VSLS production). The trace gases
include ethane ($C_2H_6$), ethyne ($C_2H_2$), propane ($C_3H_8$), n-butane ($C_4H_{10}$), benzene ($C_6H_6$), methyl chloride ($CH_3Cl$),
tetrachloroethylene ($C_2Cl_4$), methyl iodide ($CH_3I$), bromoform ($CHBr_3$), dibromomethane ($CH_2Br_2$), chloroform ($CHCl_3$), and
dichloromethane ($CH_2Cl_2$).

### 2.2.4 NOAA Water

Measurements of water vapor and total water were obtained by the NOAA hygrometer. The instrument is a two-channel,
tunable diode laser absorption spectrometer that utilizes cells that are under constant pressure, temperature, and flow conditions
(Thornberry et al. 2015). One channel measures water vapor from ambient air while the other channel measures total water
from a heated forward-facing inlet, which in the presence of clouds includes ambient water vapor plus additional water
evaporated from cloud particles. The instrument also carries an onboard calibration system to perform periodic checks on
instrument stability and performance during flight. Precision during flight was better than 0.17 ppmv (1 sec, 1 s) and
uncertainties of 5 % +/- 0.23 ppmv for water vapor and 20 % for ice water content.



### 2.2.5 MMS

Ambient physical, geophysical, and thermal parameters along the flight were measured by the Meteorological Measurement System (MMS), including pressure, temperature, and three-dimensional winds, which are calibrated and corrected for aircraft orientation (Scott et al., 1990). MMS has an integrated Global Positioning System (GPS), the LN100g INS, that provides aircraft GPS altitude, latitude, and longitude. Data are reported at a rate of 1 Hz and with a precision of 0.3 hPa for pressure and 0.3 K for temperature. Potential Temperature is calculated from MMS pressure and temperature.

Identification and characterization of the tropopause are done using MMS vertical profiles of temperature. We calculate the Lapse Rate Tropopause (LRT) based on the World Meteorological Organization definition of the lowest level where the lapse rate decreases to 2 K km$^{-1}$ or less and remains less than 2 K km$^{-1}$ in a 2-km layer above this level. We also identify the Cold Point Tropopause (CPT) as the altitude of the minimum temperature within the sampled vertical profile, provided that measurements below and above this altitude were obtained. Given the high altitude of the CPT in the tropics, the flight strategy, and aircraft limitations, the CPT was not reachable until later in each flight.

### 2.3 Satellite Observations

The Microwave Limb Sounder (MLS) was launched on the NASA Aura satellite on July 15, 2004. The instrument measures thermal microwave radiation emitted from the limb of Earth's atmosphere, forward along the satellite's flight direction. It provides global coverage from 82° N to 82° S with an Equator crossing time of 1330 local. We use the CO product retrieved from dual bands in the MLS 240 GHz radiometer (Livesey et al., 2008). Spatially, we focus on the 100 hPa level, which has a horizontal resolution of 4.5 km (cross path) by 450 km (along path) and a vertical resolution of ~4.9 km. We use daily V005 L2GP data with a reported accuracy of +/- 19 ppbv and +/- 30 % and precision of 14 ppbv. Temporally, we focus on data from middle of February to middle of March (day of year = 40 to 70), covering the period between 2010 and 2020.

### 2.4 Backward Trajectories

ATTREX gas phase in situ data are complemented with backward trajectory calculations to investigate geographical origin of air and transport processes to the TTL. The trajectory method used is described in Pfister et al., 2001, Bergman et al., 2012 and Jensen et al., 2017b. Trajectories were driven by ERA Interim temperature and wind fields with resolutions of six hours, ~1 km in the vertical in the TTL region, 0.75 degrees in the horizontal, and advected for 40 days back in time. The trajectories were run on isentropic coordinates using monthly diabatic heating rates derived from cloud observations in 2007 (Yang et al., 2010). A cluster of 25 parcels centered at the aircraft location was initialized every ~2 minutes along the flight track. The cluster was set up as a curtain perpendicular to the flight track with five heights and five trajectories at each height separated from each other within 0.5° in both latitude and longitude. In this study, we focus on trajectories initialized at aircraft altitudes only.





Once computed, trajectory temperatures were compared to 3-hourly geostationary infrared brightness temperature
and precipitation fields. Each trajectory was tagged as convectively influenced when satellite fields coincident in space and
time showed a precipitating cloud top to be at a higher altitude (lower temperature) than the parcel (Pfister et al., 2022).
Trajectories were truncated at the convective encounter closest in time to the flight date, when parcel and cloud top height
comparison criteria were met.

## 3 Results and Discussion

### 3.1 In situ Trace Gas Distributions in the TTL

Extensive high-resolution vertical sampling of the TTL was achieved during the ATTREX campaign. Figs. 2 and 3 show the
spatial distribution of $CO_2$, $CH_4$, and CO between 14 and 18.5 km in the deep tropics ($12^{\circ}$ S – $15^{\circ}$ N) over the western Pacific
in March 2014. In this study, we use GPS altitude as the vertical coordinate to evaluate trace gas and temperature variability
over time. Multiple surfaces of relevance are also shown in these figures: the LRT and the CPT in Figs. 2 and 3, and the 100
hPa isobar and the 380 K isentrope in Fig. 3. These surfaces are derived from in situ measurements of ambient temperature,
pressure, and GPS altitude. They are commonly used in TTL studies to identify and evaluate chemical, radiative, and dynamic
processes that link the tropical UT and LS and directly impact global stratospheric composition (e.g., Randel and Jensen, 2013;
Pan et al, 2018).


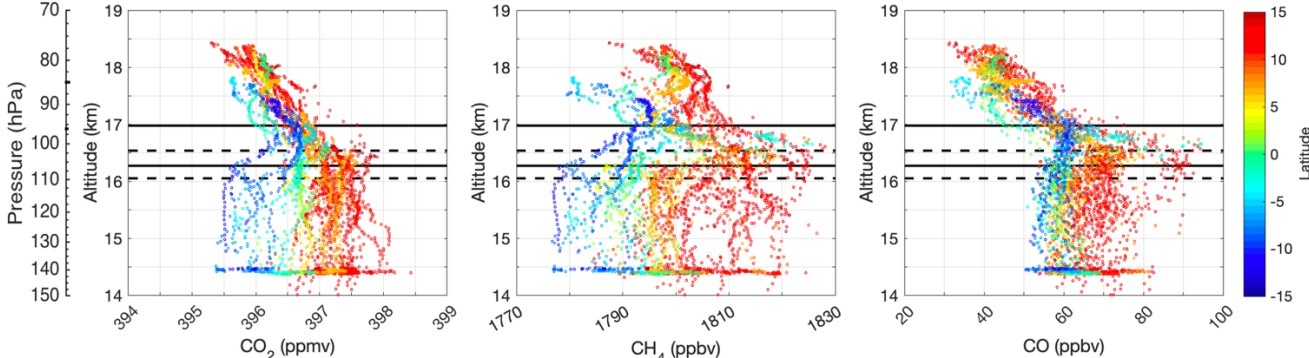

**Figure 2: ATTREX-3 vertical profiles of $CO_2$, $CH_4$ and CO in the deep tropics ($12^{\circ}$ S–$15^{\circ}$ N) over the western Pacific in March 2014 as a function of GPS altitude. Data are color-coded by latitude of sampling. Minimum and maximum Lapse Rate Tropopause heights (dashed lines) and Cold Point Tropopause heights (solid lines) are also shown.**


Several notable features stand out in the spatial distribution of these trace gases, namely: (i) large trace gas variability
within the 4-km layer sampled in the TTL and LS, (ii) interhemispheric gradients, and (iii) spatially coherent enhancements in
all trace gases up to 17 km in altitude.



The observed variability in mixing ratios provide evidence of the changing dynamic regimes as a function of altitude.
The nearly constant mixing ratios between 14.5 km and the LRT seen in Fig. 2 are consistent with convection rapidly lofting air to these altitudes. Above the CPT, the observed decreases in mixing ratios with increasing altitudes for $CH_4$ and CO are primarily driven by reaction with OH radicals. At these altitudes and low latitudes, most of the air is ascending into the tropical stratosphere as part of the global-scale Brewer-Dobson circulation, but at rates slower than the chemical reaction rates giving rise to the observed profiles. These low stratospheric altitudes in the deep tropics can also be influenced by equatorward transport and mixing in of older air. These processes would contribute to decreases in $CH_4$ and CO mixing ratios by bringing in chemically processed air with even lower $CH_4$ and CO. Contribution from these processes could be evaluated by examining additional trace gases such as $O_3$. For $CO_2$, the variability as a function of altitude is dominated by a combination of seasonal processes in the biosphere (photosynthesis and respiration), secular increases over time (e.g., Park et al, 2010), and a contribution from oxidation of $CH_4$ at higher altitudes. In between the LRT and the CPT, various physical and chemical processes are at play (Pan et al., 2018; Pan et al., 2019).

The interhemispheric gradients observed in Figs. 2 and 3 are mainly driven by emissions at the surface. These three trace gases have strong anthropogenic and natural sources. Emissions of $CO_2$ are driven by year-round fossil fuel consumption and seasonal modulations by the biosphere. Emissions of $CH_4$ are driven by natural wetlands as well as the oil and gas industry, agricultural activities, landfills, and combustion. Emissions of CO are driven by biomass burning, incomplete combustion, and as a product of oxidation of $CH_4$ and other hydrocarbons (HC). These activities vary widely between hemispheres; therefore, it is not surprising that the surface air convectively lofted to the TTL reflects such differences. Multiple measurements (ground-based, airborne, and spaceborne) have documented interhemispheric gradients of these trace gases at the surface and in the troposphere (Wofsy et al., 2011, Rigby et al., 2017, Martinez-Alonso et al., 2020). The ATTREX-3 data shown in Figs. 2 and 3, however, provide new evidence that these gradients are carried aloft into the TTL and even into the LS in the TWP, albeit attenuated in magnitude.

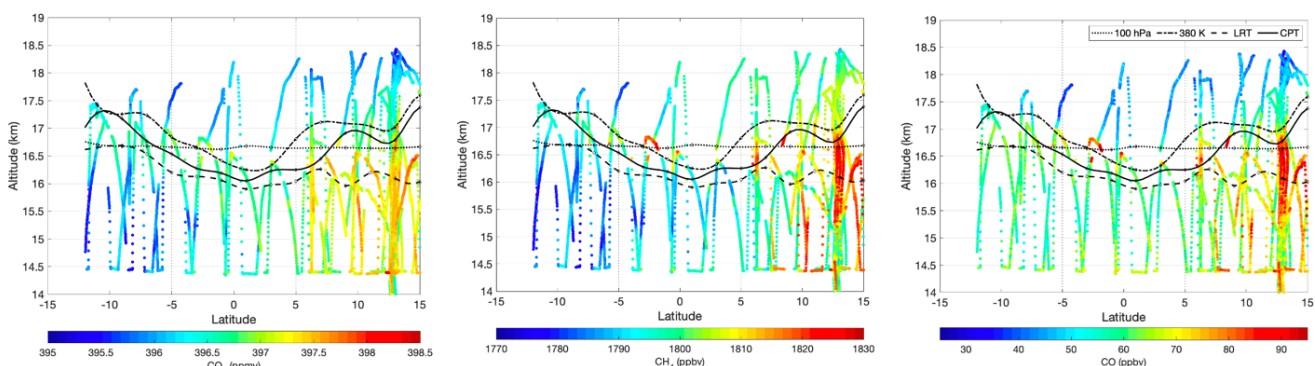

**Figure 3: Latitude distribution of $CO_2$, $CH_4$ and CO in the deep tropics ($12°S–15°N$) over the western Pacific in March 2014. Vertical coordinate is GPS altitude. Data are color-coded by trace gas mixing ratios. Black lines correspond to the Lapse Rate Tropopause**



**(LRT), Cold Point Tropopause (CPT), 100 hPa, and 380 K surfaces determined by in situ measurements of ambient temperature, pressure, and GPS altitude.**

In addition to interhemispheric differences, distinct departures from background conditions were observed throughout the TTL over the TWP. These departures were spatially coherent, especially for $CH_4$ and CO. They were observed both below

the LRT and into the LS, and as far south as 5° S as seen in Fig. 3.

Figure 4 examines the temporal evolution of the features seen in Figs. 2 and 3 focusing on vertical profiles of CO. Each panel corresponds to an individual research flight, covering the period between 20140212 and 20140311. Data are color-coded by latitude of sampling. CO mixing ratios show a maximum above 16 km in most flights as well as variability as a function of latitude and time. In February, higher CO was encountered throughout the TTL closer to the Equator (e.g.,

20140212). In March, more homogeneity within each hemisphere was observed with distinct mixing ratios regimes across the Equator on 20140309. The largest enhancements were encountered during this southern survey flight, with ratios over lower TTL mixing ratios of up to 55 % for data between 5° N and 15° N and 65 % for data 5° from the Equator. During the 20140311 flight, a difference of nearly 30 ppbv was evident in the 16-17 km layer. This difference is longitude dependent (see Fig. 1), with the higher mixing ratios sampled during the westernmost leg. Different mixing ratios suggest air masses of different origin

being advected by different flow patterns. The observation of persistent polluted layers during the period of aircraft sampling indicates that sources of elevated CO reached the TTL repeatedly.

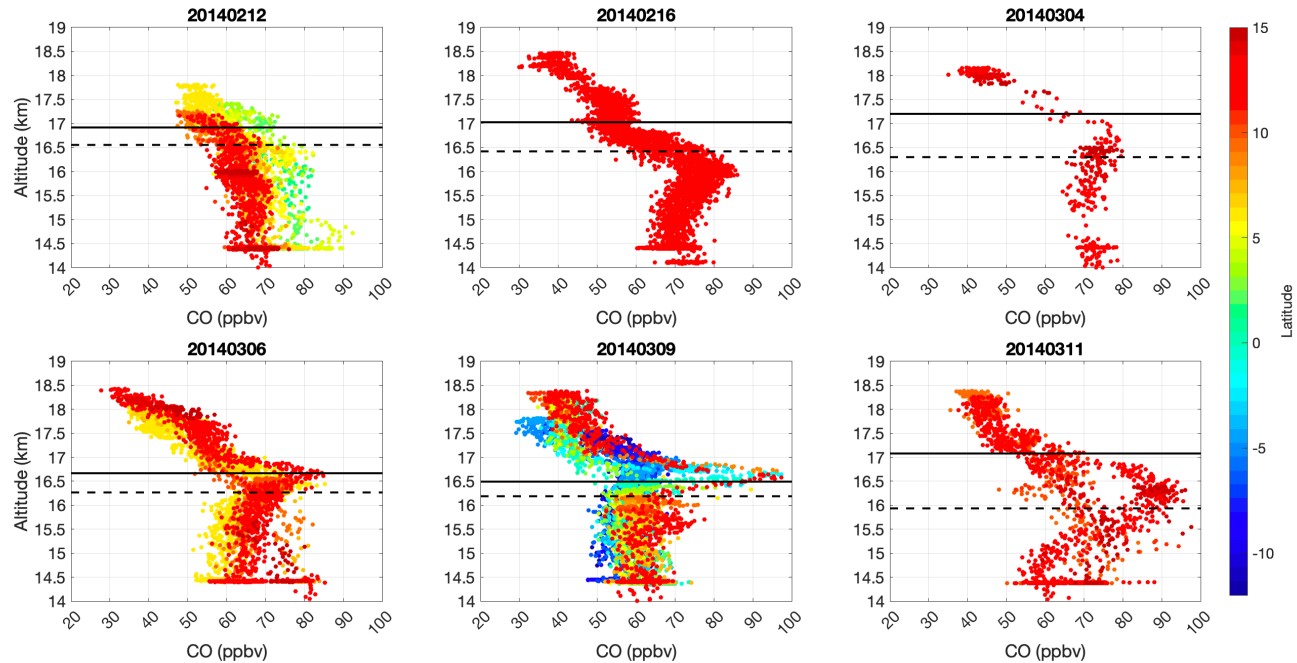



**Figure 4. ATTREX-3 vertical profiles of CO in the deep tropics (12º S–15º N) over the western Pacific in February and March 2014**
**as a function of GPS altitude. Each panel corresponds to an individual research flight with data color-coded by latitude of sampling. Also shown are the flight average Cold Point Tropopause (solid lines) and Lapse Rate Tropopause (dashed lines). The flight on 20140309 was the only one sampling the Southern Hemisphere, extending to 12º S. The observed variability between 15 and 17 km on 20140311 is driven by sampling of different air masses, separated by 13º in longitude (see flight path in Fig. 1). The higher mixing ratios were captured on the western leg of the flight.**


Guided by similarity in mixing ratios, especially as evidenced by $CO_2$, we divide the data set into three latitude bins across the Equator. These bins are defined as follows: Northern Hemisphere (NH) for data between 5º N and 15º N, Equator (EQ) for data between 5º S and 5º N, and Southern Hemisphere (SH) for data between 5º S and 12º S. Most of the sampling occurred in the NH, with EQ sampling during segments in two flights, and SH sampling during one flight only.

To further investigate the observed trace gas spatial coherence, we examine correlations of CO with $CO_2$, $CH_4$, and $O_3$ as shown in Fig. 5. This figure focuses on data north of 5º S. Data are color-coded to highlight pollution plumes at 16-17 km (red) against the LS background (blue) and the UT background (gray). The CO-$CO_2$ correlation shows distinct branches with medium levels of $CO_2$ (397 ppmv) at the highest CO. Excluding the highest CO mixing ratios, these two tracers show a strong positive correlation where younger air has higher CO and $CO_2$. This correlation is consistent with time of year, where 265 $CO_2$ in the northern hemisphere lower troposphere is heading towards peak levels as the biosphere transitions from wintertime respiration to summertime photosynthesis. The fact that the air masses with the highest CO do not have the highest $CO_2$ is an indication of unusual sources such as continental origin, a different latitude, and/or combustion origin, all factors that also contribute to $CO_2$ variability. These air masses with the highest CO and medium levels of $CO_2$ have the highest $CH_4$ mixing ratios, an indication of contribution from additional sources such as biomass burning, combustion, and the oil and gas industry. 270 Elevated CO, when accompanied by reactive HC and in the presence of nitrogen oxides, can lead to $O_3$ formation over time. During ATTREX-3, mixing ratios of $O_3$ within the CO plume remained very close to background UT levels (generally below 100 ppbv). This suggests no significant photochemical production of $O_3$ from the time of convective lofting to the time of aircraft sampling.

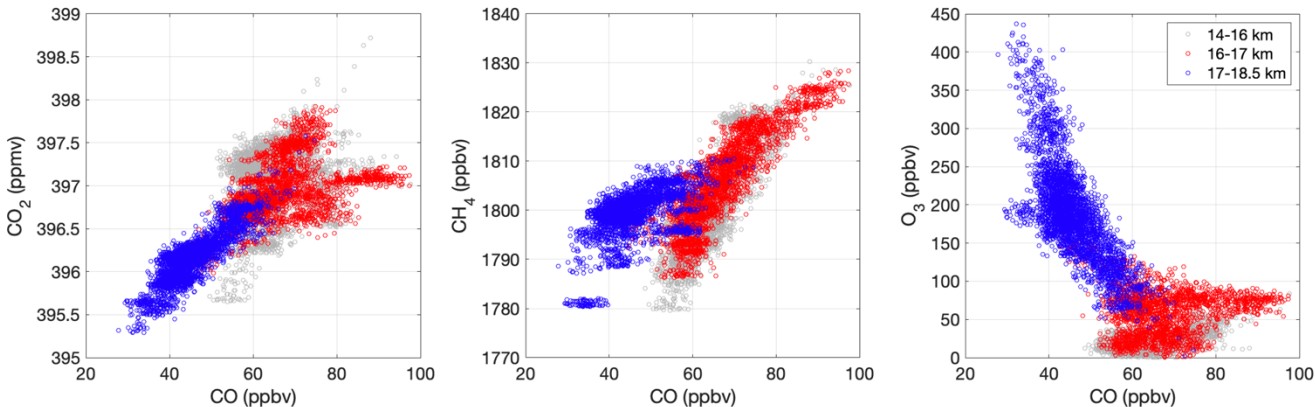






**Figure 5. Correlations of CO versus $CO_2$, $CH_4$, and $O_3$ in the deep tropics (5º S–15º N) in March 2014. Data are color-coded by altitude: below most of the pollution plumes at 14-16 km (gray), peak of the pollution layer at 16-17 km (red), and above the pollution plumes at 17-18.5 km (blue). Pollution plumes with highest CO mixing ratios (>75 ppbv at 80th percentile) are associated with distinct $CO_2$ branches, highest $CH_4$ mixing ratios, and $O_3$ mixing ratios below 100 ppbv, consistent with tropospheric values.**


### 3.2 Pollution Plume Composition

We explore the origin of pollution plumes by first examining an extensive suite of chemical trace gases. We define pollution as air masses with CO mixing ratios higher than the 80th percentile within each latitude bin. This threshold translates to 75 ppbv CO for the NH bin, 71 ppbv CO for the EQ bin, and 63 ppbv CO for the SH bin.

Figure 6 shows vertical profiles of CO, $CH_4$, and HCs with varying atmospheric lifetimes: $C_3H_8$ (2 weeks), $C_2H_2$ (3 weeks), $CH_3Cl$ (1.5 years), and $C_2Cl_4$ (4 months), collected in March 2014. Data are grouped in the three latitude bins defined above. The extreme mixing ratios observed by HUPCRS were not captured in the GWAS data as shown in this figure. Pairing of these two instruments with significantly different response times required averaging HUPCRS data to GWAS sampling times, which were increasingly longer at higher altitudes, where we observed peak mixing ratios for multiple trace gases.

Pairing of these two instruments also contributed to a loss of up to 20 % of the data, because some of the GWAS sampling occurred during periodic in-flight calibration times for HUPCRS. Despite these limitations, there is clear evidence of the presence of these unexpected pollutants in the TTL and LS over the TWP, including gases with very short atmospheric lifetimes such as propane and ethyne, in both the northern hemisphere and the equatorial zone.



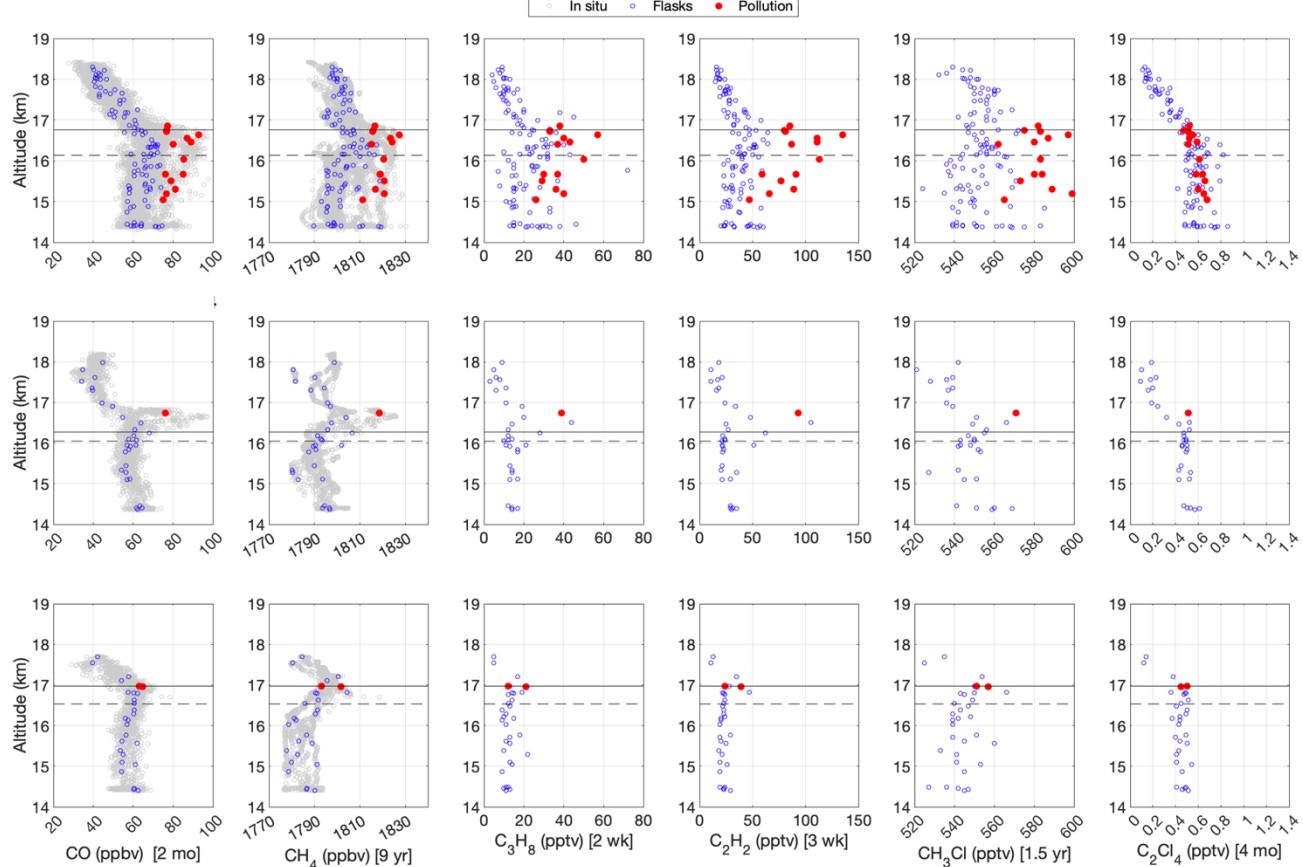


**Figure 6. Vertical profiles of CO, CH₄, C₃H₈, C₂H₂, CH₃Cl, C₂Cl₄ in the deep tropics over the western Pacific in March 2014. Gray points are for in situ measurements, blue points are for GWAS canisters, and red points are GWAS canisters with CO mixing ratios above the 80th percentile within each latitude bin. Also shown are the latitudinally averaged Cold Point Tropopause (solid) and Lapse Rate Tropopause (dash). Top row are NH bin data (5°-15° N), middle row are EQ bin data (5° S-5° N), and bottom row are SH bin**
**data (5°-12° S).**

The HCs and CH₃Cl shown in Fig. 6 are known to be emitted by biomass burning (Mauzerall et al., 1998; Blake et al., 1999; Akagi et al., 2011; Santee et al., 2013; Andreae, 2019). In all instances, HCs and CH₃Cl were positively correlated with CO and CH₄, but C₂Cl₄, a tracer of urban origin, remained unchanged. This result points to biomass burning as the source
of these pollution plumes, with undetectable urban/industrial contributions. The HC data also confirm that pollution inputs were pervasive for the duration of the campaign and extended across the Equator, well into the thermally defined LS.

We evaluate enhancement ratios using a least-square regression of CO versus trace gases with a wide range of atmospheric lifetimes, ranging from a few days (e.g., n-C₄H₁₀) to months (e.g., C₂H₆) and years (e.g., CH₃Cl and CH₄). The enhancement ratio (slope) is obtained using the reduced-major-axis method (Hirsch and Gilroy, 1984). In addition to slopes,
we calculate the coefficient of determination (r²) and p-values at the 95 % confidence interval to assess the statistical



significance of the correlations. Figure 7 shows trace gas correlations with CO and $CH_4$, focusing on altitudes below 17 km. The color-coded data are for the southern survey flight only (20140309), with the NH bin represented by filled circles and the EQ bin represented by filled diamonds. The least-square regression is performed on a single flight (20140309) with data below 17 km in order to exclude flight-to-flight variability in UT background mixing ratios (as seen in Figs. 4 and 6) and

photochemically aged air at higher altitudes.

All trace gas pairings with CO show strong and positive correlations in the NH bin. Lower $r^2$ values are observed for trace gases with lifetimes shorter than 10 days, and all other trace gases show $r^2$ over 0.9. In all cases, p-values are less than 0.05 (0.0129 for n-$C_4H_{10}$, 0.0017 for $C_6H_6$, and less than 0.00001 for all others), which indicates that the correlations are statistically significant. Given the limited number of canisters in the EQ bin, we do not perform a statistical analysis; however,

we note similar patterns qualitatively, especially for trace gases with atmospheric lifetimes longer than 10 days. The magnitudes of the slopes obtained are consistent with previous aircraft studies of biomass burning, except for correlations of $CH_4$ with $C_2H_6$ and CO, where slopes were nearly an order of magnitude larger during ATTREX-3 (Mauzerall et al, 1998; Mühle et al, 2002; Gkatzelis et al., 2023). The observed excess $CH_4$ suggests influences of additional sources such as emissions from the oil and gas industry (Mühle et al, 2002).

We consider a wide range of atmospheric lifetimes to assess an approximate transport timescale based on which set of trace gases exhibits weaker correlations with CO. Weaker correlations arise, in part, due to faster chemical loss compared to transport rate. On 20140309, we find correlations to be weakest for trace gases with atmospheric lifetimes shorter than 10 days, suggesting that it took at least that amount of time for the polluted air to reach our region of sampling. These results are in agreement with previous measurements (Mauzerall et al., 1998; Gkatzelis et al., 2023).




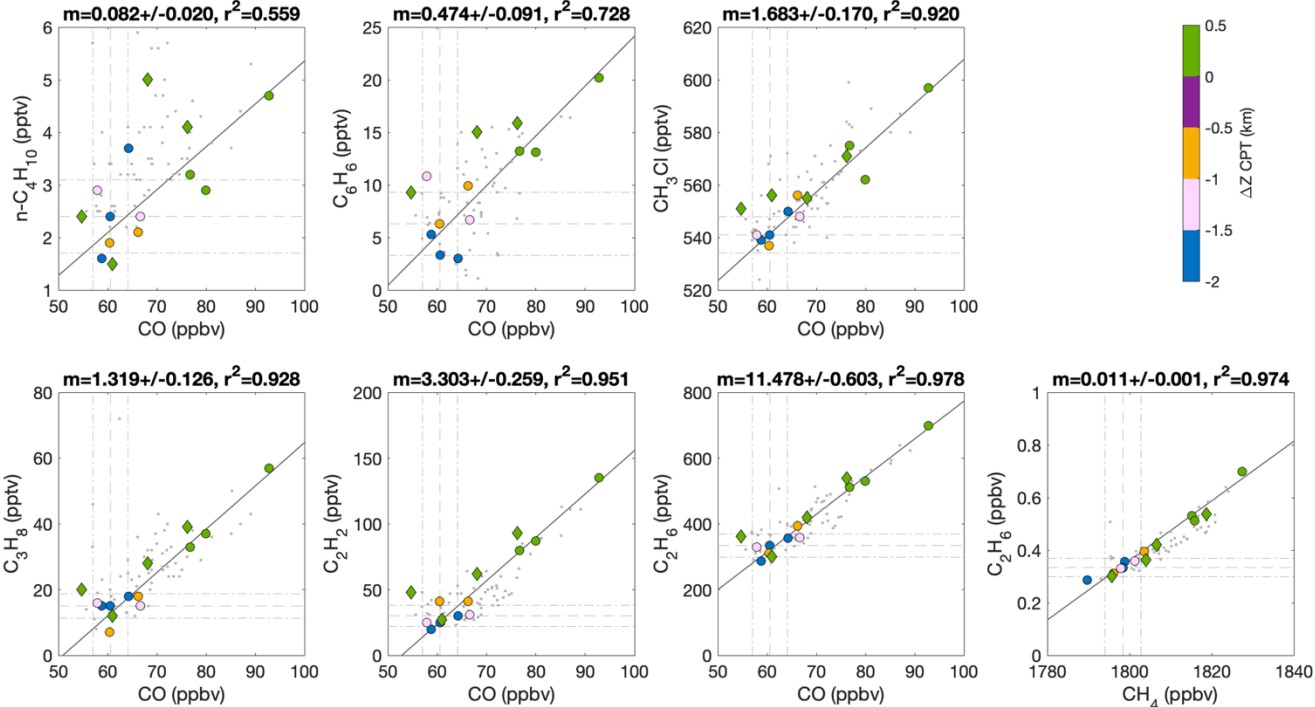

**Figure 7. Correlations of CO versus various hydrocarbons and CH$_3$Cl, as well as CH$_4$ versus C$_2$H$_6$. Data are between 14 and 17 km. Gray open circles correspond to March flights between 20140304 and 20140311. Color-coded data are for the NH bin (circles) and the EQ bin (diamond) on 20140309, where the color corresponds to the vertical distance to the Cold Point Tropopause. The latitude**
**bins are defined as 5$^o$-15$^o$ N for the NH bin and 5$^o$ S-5$^o$ N for the EQ bin.**

        Next, we examine various halogenated VSLS associated with the pollution plume. Figure 8 is the same as Fig. 6, but for vertical profiles of CH$_3$I, CHBr$_3$, CH$_2$Br$_2$, CHCl$_3$, and CH$_2$Cl$_2$ along with CO. The Chlorine VSLS are chosen as markers of anthropogenic emissions from urban or industrial source regions, and Bromine and Iodine VSLS reflect oceanic sources,
though significant anthropogenic emissions of Bromine VSLS in coastal regions have been reported (Jia et al., 2023). In all cases, we find no changes in these halogenated compounds within the pollution air compared to background levels. The only enhancement observed was in CH$_3$I, but not within the pollution plume. On 20140304, we sampled recent convective air (0.5 to 2 days old) in the TTL, just south of the active Typhoon Faxai (Jensen et al., 2017a). The rapid injection of surface air to the TTL was evident in the elevated CH$_3$I mixing ratios between 16 and 17 km. Longer lived trace gases such as CO$_2$ and CH$_4$
showed mixing ratios consistent with nearby ground stations during the timeframe of the storm, corroborating the age of the air (Jensen et al., 2017a). The observed mixing ratios of the halogenated VSLS shown in Fig. 8 confirm undetectable influence of urban sources or oceanic emissions, hence no transport of excess halogenated VSLS to the TWP by these pollution plumes. Furthermore, the low levels of CHBr$_3$ in the pollution plumes are consistent with a continental source that is not influenced by any nearby coastal or oceanic emissions.






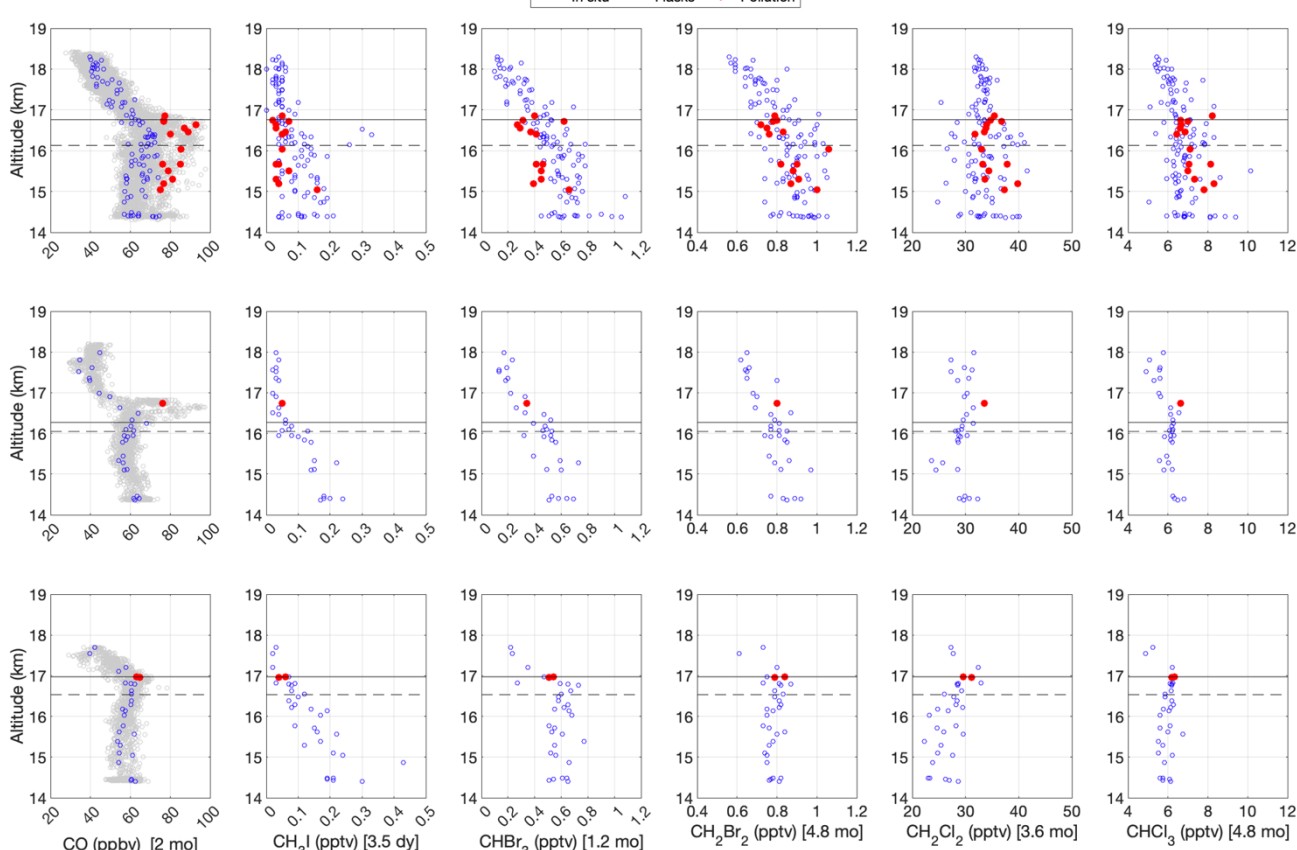

**Figure 8. Vertical profiles of CO, CH₃I, Bromine VSLS, and Chlorine VSLS in the deep tropics over the western Pacific in March 2014. Gray points are for in situ measurements, blue points are for GWAS canisters, and red points are GWAS canisters with CO mixing ratios above the 80th percentile within each latitude bin. Also shown are the latitudinally averaged Cold Point Tropopause (solid) and Lapse Rate Tropopause (dash). Top row are NH bin data (5º-15º N), middle row are EQ bin data (5º S-5º N), and bottom row are SH bin data (5º-12º S).**

In addition to hydrocarbons and halocarbons, it is also worth exploring the relative humidity conditions in the pollution plumes. Figure 9 shows the correlation between CO and relative humidity with respect to ice in clear air and in clouds within the 16-17 km layer in March 2014. We choose a threshold for clouds of enhanced total water greater than 5 ppmv over water vapor. This choice allows an examination of the more frequently encountered thicker clouds. A lower threshold for cloud detection of 1 ppmv, for instance, only impacts 2 % of the data within the pollution plumes and has no effect on the overall conclusions. We found the air masses within the pollution plume, defined as CO mixing ratios above the 80th percentile, to be mostly in clear air (88 %) compared to in clouds (12 %), and mostly subsaturated (76 %) compared to saturated (8 %) and supersaturated (15 %). We define saturation as relative humidity of 98-102 % and supersaturation as relative humidity above



102 %. These ranges and thresholds were chosen to account for measurement uncertainties and variability in ambient conditions (Rollins et al., 2016; Jensen et al., 2017b).

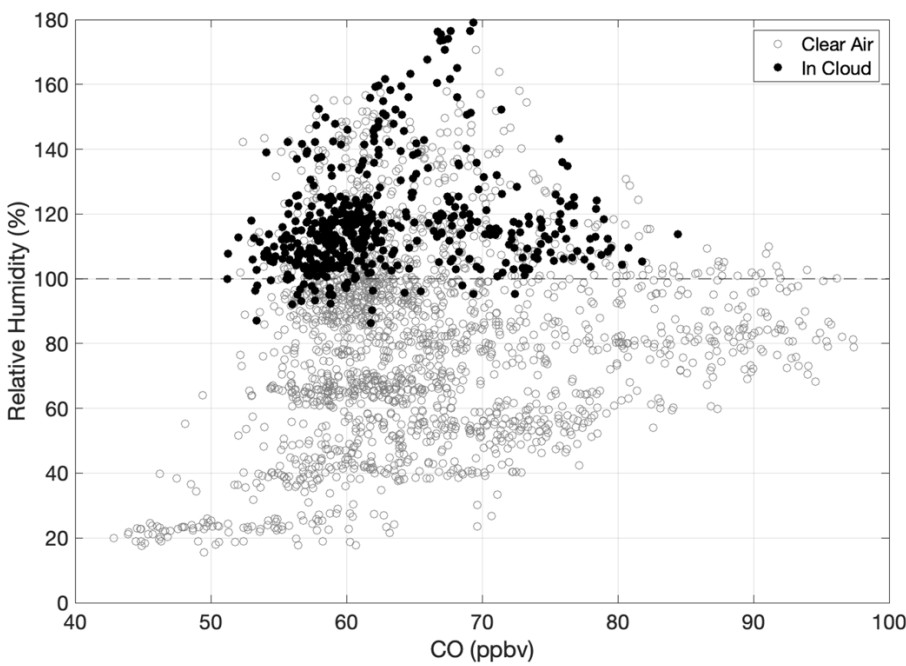

**Figure 9. Correlation of CO and relative humidity over ice for air masses sampled in the deep tropics (12º S–15º N) over the tropical**
**western Pacific in March 2014. Data are at altitudes of peak CO mixing ratios, between 16 and 17 km. Solid circles correspond to air masses inside clouds, with clouds defined as enhanced total water greater than 5 ppmv over water vapor. Open circles correspond to cloud-free air.**

Ambient temperature and $H_2O$ varied within the pollution layer. Figure 10 illustrates their correlation in clear air and
in clouds within the context of CO mixing ratios. The data shown include all air masses in the 16-17 km layer during March 2014, regardless of CO mixing ratios. Also included in the figure are lines of constant relative humidity over ice. Relative humidity inside clouds closely followed the 100 % line in most cases, with a few departures mainly towards supersaturation conditions. Most of the air inside clouds was not impacted by pollution, leaving evidence of pollution primarily in cloud free air. Three distinct populations stand out in the cloud-free regime. The first regime is air masses with $H_2O$ below 3.5 ppmv,
which were encountered over a wide range of temperatures (187.5 – 201 K) and CO mixing ratios (42.8 – 97.4 ppbv). Most of these low $H_2O$ air masses were in subsaturated air, an indication that local temperature conditions were not the drivers for dehydration to the observed levels, consistent with previous studies (Gettelman et al., 2002b; Fueglistaler et al, 2004; Pan et al., 2019). No correlation between temperature and CO was observed at these low $H_2O$ levels. The second regime is air masses with $H_2O$ above 5 ppmv, which were encountered over a narrower range of temperatures (190.6 – 192.6 K) and CO mixing ratios (57.2 – 74.8 ppbv). These air masses were all in supersaturated air. The third regime is air masses with $H_2O$ between 3.5



and 5 ppmv, which were encountered over a wider range of temperatures (189.4 – 194.9 K) and CO mixing ratios (52.1 – 95.5 ppbv) compared to the second regime. Unlike the other two regimes, this third regime shows a strong correlation between relative humidity and CO mixing ratios where pollution plumes were found in warmer and subsaturated air. Histograms of ambient temperature in clean versus polluted air masses in this third regime ($H_2O$ between 3.5 and 5 ppmv) are shown in Fig.

11. These histograms allow us to identify and quantify the shift to warmer temperatures observed within the pollution plumes. On average, we find a temperature increase of 1.52 K in air masses with the more extreme pollution levels (90[th] percentile CO). These warmer and polluted air masses were encountered 15 % of the time in the 3.5 to 5 ppmv $H_2O$ range and at altitudes between 16 and 17 km.

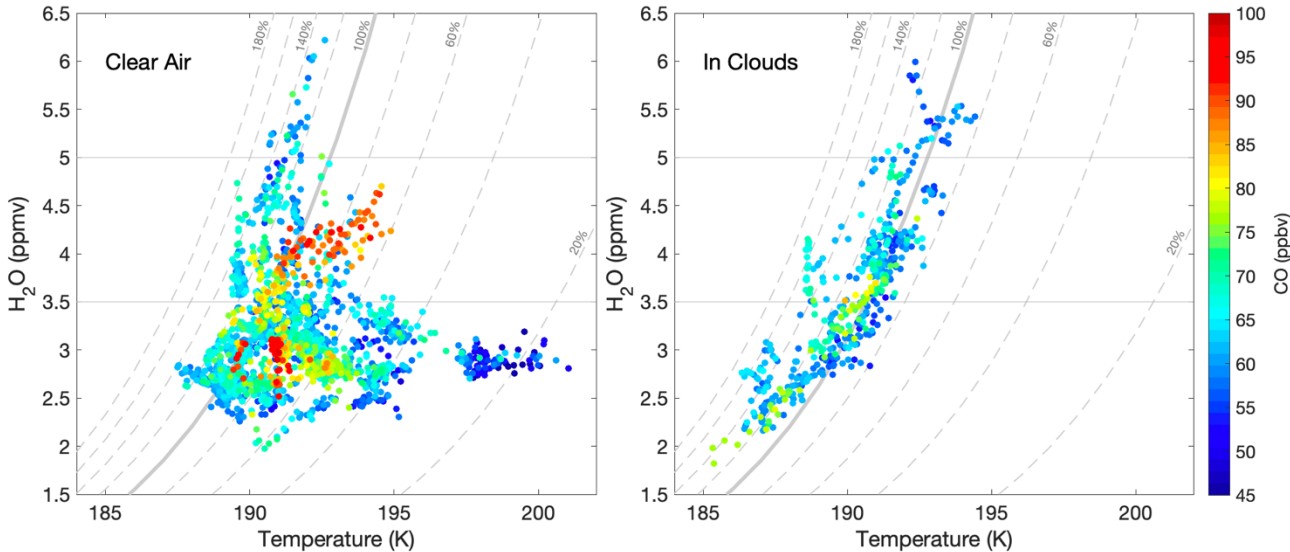


**Figure 10. Correlation of ambient temperature and water vapor for air masses sampled in the deep tropics (12º S–15º N) over the tropical western Pacific in March 2014. Data are at altitudes of peak CO mixing ratios, between 16 and 17 km, and color-coded by CO mixing ratios. Lines of constant relative humidity over ice are also shown (solid for 100% and dashed for all others). Left panel**
**is for clear air and right panel is inside clouds. Clouds are defined as enhanced total water greater than 5 ppmv over water vapor. Three water vapor regimes are analyzed in this study: below 3.5 ppmv, between 3.5 and 5 ppmv, and above 5 ppmv. See text for details.**



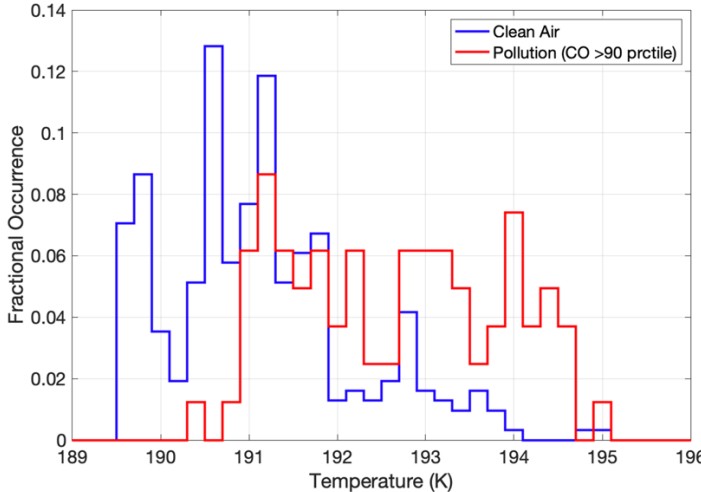

**Figure 11. Histograms of ambient temperature in cloud-free air at water vapor mixing ratios between 3.5 and 5 ppmv and altitudes**
**between 16 and 17 km. A 90$^{th}$ percentile threshold in CO is used to separate polluted air (red) from clean air (blue). These air masses**
**were sampled in the deep tropics (12$^o$ S–15$^o$ N) over the tropical western Pacific in March 2014.**

### 3.3 Satellite Observations of CO

In this section, we examine the larger spatial and temporal context of the pollution plumes observed during ATTREX-3 using
100-hPa MLS CO. Figure 12 shows data between 15$^o$ S and 15$^o$ N during February and March 2014. The in situ data shown
are for the layer between 80 and 140 hPa. These altitudes roughly correspond to the weighted levels included in the 100-hPa
MLS retrieval. All individual tropical MLS observations as well as statistics (i.e., 10-degree longitude binned average and 2
standard deviations) are also shown. We find the range in CO to be comparable between the two datasets. Out of all longitudes,
Africa stands out as the location of highest CO, both on average and extreme values. In terms of latitudinal distribution, we
find CO hot spots across the Equator over both Africa and Indonesia and in the NH over the central Pacific, as shown in Fig.
13. Previous studies have reported CO hot spots over the same geographical regions during boreal winter over different years,
indicating that these are persistent seasonal sources of elevated CO (Schoeberl et al, 2006; Duncan et al., 2007; Huang et al,
2012). A comparison of the 2014 ATTREX campaign to the MLS period from 2010 to 2020 showed that 2014 was at or below
average for extreme CO mixing ratios across the tropics (see Supplement section, Fig. S4).



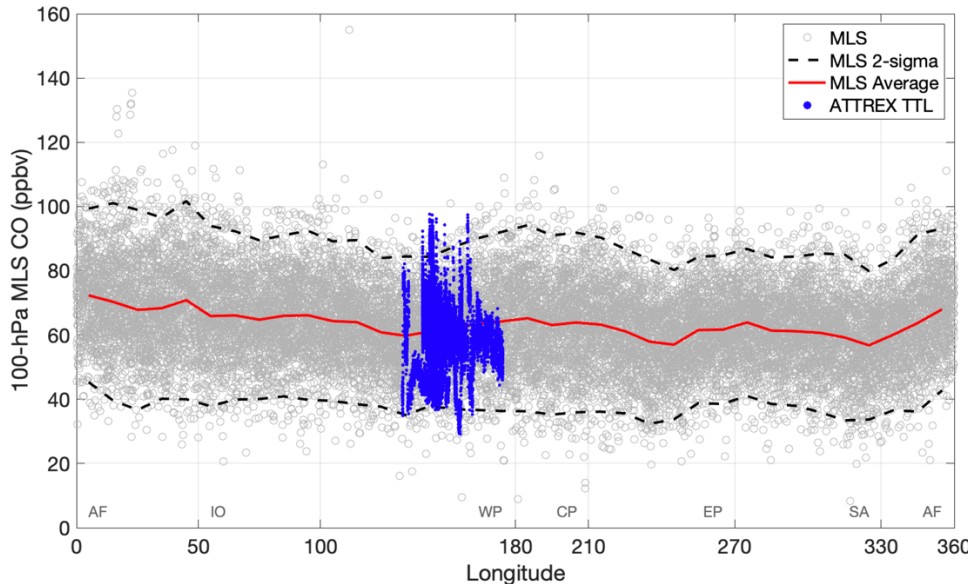

**Figure 12. Aircraft (blue) and 100-hPa satellite observations from MLS (gray) of CO across the deep tropics (15º S–15º N) in February-March 2014 (Julian Day = 40-70). The aircraft data shown are between 80 and 140 hPa, the layer equivalent to the vertical resolution of the 100-hPa MLS CO retrieval.**

Comparing in situ aircraft data and spaceborne observations requires a careful approach, especially when exploring extreme values. Spatial and temporal coincidences as well as favorable environmental conditions for the ideal comparisons are nearly impossible to achieve. The more sensible approach is to explore the statistics of the measurements considering larger areas of measurements. One consideration in this comparison approach is vertical resolution of the datasets. In situ CO shows the pollution layer to be contained mainly between 15.5 and 17 km (see Figs. 2-4). The closest altitude in the MLS retrieval is the 100-hPa level, which comprises a weighted sampling ~4.9 km in depth that extends hundreds of kilometers horizontally. This coarser spatial resolution makes it challenging to detect extreme events from space when they occupy only a fraction of the retrieved volume. A second consideration is the uncertainty of the measurement. Even though the CO range is comparable between the aircraft and the satellite instruments for data over the TWP, a Kolmogorov-Smirnov test at the 95 % confidence interval revealed the satellite data to not be statistically different from a normal distribution. In contrast, the aircraft data did not follow a normal distribution (see Supplement section, Fig. S5). These results imply that the extreme high values in the aircraft measurements are signal, but they are indistinguishable from noise in the spaceborne observations obtained in the vicinity (space and time) of the aircraft sampling.





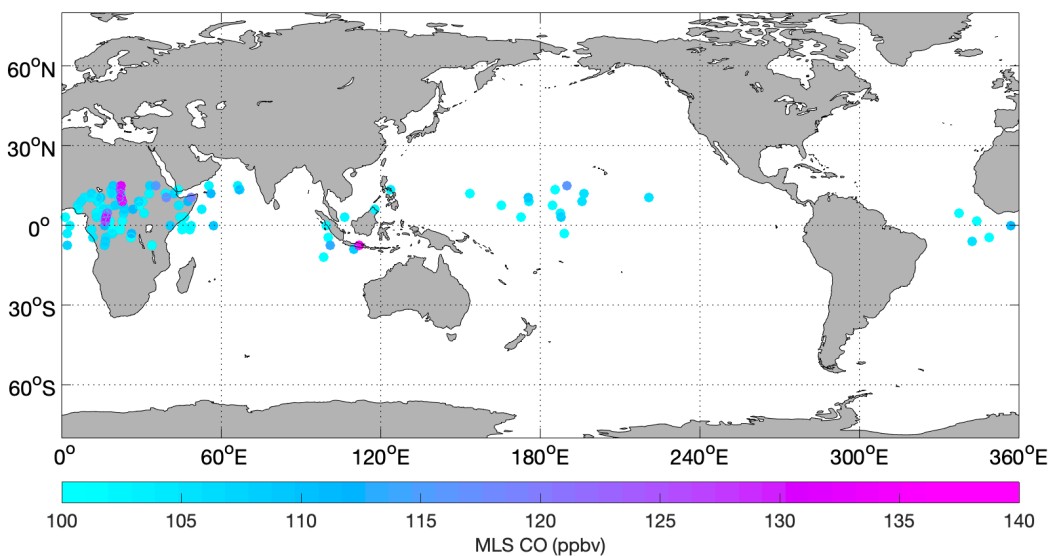

**Figure 13. Satellite observations of CO at 100 hPa obtained by the Microwave Limb Sounder (MLS) across the deep tropics (15º S–15º N) in February-March 2014 (Julian Day = 40-70). Data are for CO>100 ppbv. The largest mixing ratio of 155 ppbv is found over Indonesia. This single value is excluded from the color bar to better visualize the range of the remaining observations. These high CO values were found over tropical Africa, Indonesia, and the western/central Pacific.**

**3.4 Convective Origin and Transport Timescales from Backward Trajectories**

Convection is responsible for lofting surface into the TTL (Fueglistaler et al., 2009 and references therein). We investigate the geographical origin, transport pathway, and advection timescales of the pollution plumes by calculating 40-day backward trajectories initialized along the flight path and terminated over regions of most recent convective encounter. Figure 14 shows the convective properties of the pollution plumes sampled on 20140309, north of 5º S, and between 16 and 17 km.





**Figure 14. Convective properties of polluted air sampled in the 16 - 17 km layer between 5º S and 15º N on 20140309 based on backward trajectory analysis. Pollution is defined as CO mixing ratios above the 80$^{th}$ percentile within the latitude bin (5º S-5º N and 5º-15º N). (a) Transport pathways between location of convective encounter determined by trajectory calculations and aircraft**
**sampling. (b) Histograms of transport timescales and (c) time series of trajectory height for the dominant convective source regions. (d) Height (in hPa) of the aircraft sampling (black) and convective cloud tops over the dominant source regions. The color-coding in (a) and (d) corresponds to number of days between convective encounter by the trajectories and aircraft sampling. The black rectangle and black diamonds shown on the map in (a) correspond to the location of ATTREX-3 measurements and convective encounters, respectively.**



The trajectories of the pollution plumes sampled in both the NH and EQ bins followed the same general transport pattern, namely tropical convective lofting followed by eastward advection at subtropical latitudes along the jet, then anticyclonic transport over the western Pacific to the location of aircraft sampling. In some cases, westward advection along equatorial latitudes preceded the eastward transport along the jet. Convective encounter occurred over three distinct tropical regions: western/central Pacific, Africa, and Indonesia. The geographical contributions within the pollution plumes were as

follows: 34.5 % from Africa, 25.9 % from Indonesia, 29.3 % from TWP, and 5.2 % from the central Pacific. These percentages correspond to 60.4 % contribution from continental convection and 34.5 % contribution from marine convection.

In general, the transport timescales were proportional to the longitudinal separation between areas of convection and aircraft sampling. The shortest transport timescales were for air masses from marine convection nearby, 10 days prior to sampling. There were a few instances, however, when the nearby marine convection occurred five weeks prior to aircraft

sampling. Air masses associated with convection over land travelled for longer periods of times, two to four weeks from Africa and four weeks from Indonesia. Even though Indonesia is closer to the TWP, air masses lofted by local convection travelled westward first due to dominant upper-level easterly winds at these low latitudes.

Trajectory analysis indicated that convective cloud tops associated with the pollution plumes reached the bottom and middle of the TTL and remained within the TTL as they gradually ascended to the altitudes of the aircraft sampling (see Fig.

14 (c) - (d)). In some cases, however, convective cloud tops reached the upper TTL. These cases occurred mainly over Africa. These results are consistent with satellite observations showing that the largest percentage of deep convection penetrating the TTL occurs over tropical Africa during boreal winter (Alcala and Dessler, 2002; Liu et al., 2020).

Strong fire activity was evident from space over equatorial Africa and Indonesia in February and March 2014 (Anderson et al., 2016). Associating the observed elevated CO, $CH_4$, and various HCs with biomass burning over Africa and

Indonesia is therefore very plausible.

If instead we attribute pollution plumes with convection over the TWP and central Pacific, we require additional considerations. The polluted air masses are clearly not of marine, but of continental origin based on their chemical composition. One possibility would be a different advection-convection transport pattern. If convection is lofting air from the surface below, polluted air would first need to be advected from continental sources to the TWP following a general eastward flow at lower

altitudes prior to convective entrainment and lofting over the TWP. These various combinations of convection and long-range advection transport processes have been identified in previous studies (Jiang et al, 2007; Huang et al., 2012), and thus cannot be ruled out. Another consideration would be wind uncertainties in operational analyses. The analyses rely on observations such as those from radiosonde stations. In remote regions like the western Pacific, limited observations exist, which contributes to an increase in meteorological field uncertainties (Podglajen et al., 2014).

Results from these trajectory calculations are consistent with the chemical composition analysis described in Section 3.2 in terms of geographic source regions and transport timescales. Analysis of aircraft measurements collected over nearly a decade revealed not only that Africa was a persistent source of CO in the UT, but that anthropogenic sources and biomass burning contributed the same order of magnitude to the UT CO budget during boreal winter (Lannuque et al., 2021). This





finding could explain the discrepancy between observed $CH_4$ enhancements during ATTREX-3 and those expected from a
biomass burning source alone by attributing the difference to other sources of pollution to the UT such as the oil and gas
industry, for instance.

## 4 Conclusions

In situ and flask measurements of multiple trace gases revealed frequent, horizontally wide-spread, and vertically compact
pollution plumes of continental origin reaching the Tropical Tropopause Layer and Lower Stratosphere over the tropical
western Pacific upwelling region during the boreal winter of 2014.

Analysis of the chemical composition of the pollution plumes using correlations of multiple trace gases including
$CO$, $CH_4$, $CO_2$, $O_3$, $H_2O$, HCs and halocarbons indicated biomass burning as the dominant source of pollution. A possible
contribution from the oil and gas industry could not be excluded. No enhancements in halogenated VSLS were found in these
plumes, suggesting undetectable contributions from urban areas or oceans (e.g., $CHBr_3$). These pollution plumes were also
found in warmer air masses of equivalent $H_2O$ mixing ratios, and hence lower relative humidities up to 40% below saturation.
Additional data and further analysis would be needed to assess whether an underlying physical process exists linking the
presence of biomass burning products (e.g., extreme $CO$ mixing ratios in our case) with local warmer temperatures and
suppression of cloud formation, the latter which is a key mechanism in the TTL for regulating water vapor mixing ratios
entering the global stratosphere (Randel and Jensen, 2013).

Satellite observations of $CO$ from MLS along with backward trajectory calculations coupled with convective
observations were in agreement with geographical source regions for these plumes, namely Africa, Indonesia, and
western/central Pacific. Continental contribution from Africa and Indonesia was dominant at 60.4 % compared to a 34.5 %
contribution from the western/central Pacific. A ten-year MLS record showed that Africa is a consistent source region for TTL
$CO$ in the deep tropics during boreal winter and that 2014, the year of aircraft measurements analyzed in this study, was below
average for extreme $CO$ mixing ratios across the deep tropics. The high $CO$ events measured by the aircraft over the TWP
were not detected by MLS. Possible explanations for the discrepancy include different volumes of air sampled by each
platform, different vertical resolution, and instrument sensitivity to sporadic high mixing ratios that are comparable in
magnitude to the noise in the retrieval.

Backward trajectory calculations also provided transit times between convective delivery and aircraft sampling in the
TTL and LS over the TWP. Transport timescales from continental convection ranged between two and four weeks, while a
wider range of one to five weeks was associated with transport from nearby marine convection. These transport timescales
were consistent with atmospheric lifetimes of the trace gases examined.

Satellite observations and trajectory studies have identified the TWP as the main region where convectively lofted air
to the TTL enters the tropical stratosphere, with fastest ascent rates through the TTL during boreal winter. This study,
conducted over those longitudes and during that time of year, reveals that air sampled in the TTL and LS was sourced not only





from the nearby marine boundary layer, but also from distant, continental regions across the tropics prior to convective lofting to the TTL and advection to the TWP.

Stratospheric composition, which has a direct impact on the recovery of the $O_3$ layer, depends on the composition of tropospheric air convectively lofted to the TTL. Of particular concern is the fast convective delivery of halogenated VSLS to
the TTL and tropical LS, where their short atmospheric lifetimes are no longer a limiting factor influencing stratospheric $O_3$ concentrations. Rapid delivery of halogenated VSLS from Asian emission sources to the TTL over the TWP during boreal autumn, a season of slower ascent into the stratosphere, have been reported (Treadaway et al., 2022). The present study did not find evidence for VSLS transport to the TWP, however.

This study did find rapid delivery of trace gases from biomass burning sources, including gas phase precursors to
secondary organic aerosols such as $C_6H_6$ (Borrás and Tortajada-Genaro, 2012; Arias et al., 2021). The associated aerosols have been hypothesized to contribute to stratospheric $O_3$ loss (Solomon et al., 2023) via halogen activation under new thermal and chemical environment. Biomass burning inputs can also impact the radiative properties of the TTL by altering concentrations of radiatively active gases such as $CH_4$, $CO_2$, and $O_3$ (Gettelman et al, 2004) and light scattering and absorption by aerosols, which modify scattering and absorption of shortwave and longwave radiation as well as influence cloud formation and life
cycles (Fueglistaler et al, 2009; Huynh and McNeill, 2024; and references therein). An increase in intensity and frequency of fire activity in a warming climate coupled with biomass burning plumes rapidly reaching the TWP, as shown in this study, could have a significant impact on both radiation and chemistry in the TTL and tropical LS regions and ultimately on global stratospheric composition and climate.

Multiple studies have now shown that a variety of source regions of pollution can rapidly reach the remote and critical
altitudes of the TTL and LS over the TWP year-round. Accurately forecasting changes in stratospheric composition and climate hence hinges on continuously monitoring the composition of the atmosphere at multiple spatial and temporal scales.

*Data availability.* NASA ATTREX data are freely accessible from the NASA ESPO archive at https://espoarchive.nasa.gov/archive/browse/attrex/id4. Likewise, NASA Aura MLS data can be freely obtained from https://disc.gsfc.nasa.gov/datasets?page=1&keywords=AURA%20MLS.

*Supplement link.*


*Author contributions.* JP and SW designed and executed the analysis. All co-authors performed and QA/QC the measurements and calculations. JP drafted the manuscript and co-authors provided editorial comments and corrections.

*Competing interests. The authors declare that they have no conflict of interest.*




*Acknowledgements.* This work was supported by the NASA Grants NNX10AO82A and NNA15BB89P. The authors would like to gratefully acknowledge the pilots and ground crew of the NASA Global Hawk aircraft as well as the NASA ESPO team for their dedication and great work. We would also like to thank the JPL MLS science team for providing satellite data. We wish to extend a special thank you to M. Sargent and J. Lindaas for their invaluable contribution to field operations. Elliot
Atlas acknowledges support by NASA Grant NNX10AO83A and contributions of R. Lueb, R. Hendershot, and S. Gabbard for technical support in the field, and X. Zhu and L. Pope for GWAS lab analysis.

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
