# Peer review of "Aircraft Observations of Biomass Burning Pollutants in the Equatorial Lower Stratosphere over the Tropical Western Pacific During Boreal Winter"

_EGUsphere, 2024_

## Author Response (AR1)

**RESPONSE TO REVIEWERS FOR EGUSPHERE PAPER #2024-3832**

We would like to thank the reviewers for valuable comments that helped improve and strengthen our manuscript. Our responses are shown in *green*.

**ACPD – Referee #1 Comments**

**RC1**: 'Comment on egusphere-2024-3832', Anonymous Referee #1, 13 Jan 2025  reply

The manuscript "Aircraft Observations of Continental Pollution In the Equatorial Lower Stratosphere over the Tropical Western Pacific During Boreal Winter" by Pittmann et al. presents observations from the ATTREX campaign 2014 over the tropical western Pacific (TWP) region obtained with the Global Hawk.

The paper describes the dataset and comes to the conclusion that the UTLS composition in the TWP is dominated during the campaign time by continental pollution.

In my point of view the paper is in general well written, but some major and minor revisions should be conducted for a final publication in ACP.

1) The pollution is termed "continental pollution". Even though the analysed pollution is of continental origin, the authors state that this is mostly from biomass burning pollution. Consequently, a re-naming of the manuscript should be considered, as the influence of anthropogenic pollution in the TWP UTLS is minor.
*We appreciate the suggestion. We replaced the term "Continental Pollution" with "Biomass Burning Pollutants" in the title.*

2) The authors make both in the abstract as well as in the conclusions statements about aerosol particles. However, the measurement data does not obtain any data point on aerosol particles. Even though aerosol particles are potentially included in the emission sources, their fate and transformation in the convective ascent as well as microphysical processing and / or sedimentation along the relatively long transit times is uncertain. I would expect substantial scavenging of primary emitted particles in the ascent, either by precipitation scavenging or by ice nucleation such that only a minor fraction of primary aerosol particles will be transported into the UTLS. Whether new particle formation also takes place in the TWP or in the transit to the measurement region is not yet certain. Therefore, the TWP cannot be named an ascent region of aerosols into the LS. And as this is not shown in this manuscript (and it is not its intention), these statements should be either weakened and discussed or should be removed from the manuscript.

Similarly, the role of VSLS are prominently mentioned in the beginning, but the analysis

showed that they do not play a significant role in this context, i.e., as lofted species with a high ozone destruction potential.

*Aerosols and VSLS are important players in the recovery of the stratospheric $O_3$ layer, so they provide motivation for the analysis we conducted.*
*In terms of aerosols, it is indeed impossible to determine if the pollution plumes we sampled carried the expected emitted aerosols along the way, especially over multiple days and long distances, given the absence of measurements. The only data we have are of Secondary Organic Aerosol (SOA) precursors, such as Benzene. Elevated values of this trace gas suggest an increased potential for SOA formation, which is all we can deduce. Our study confirms delivery of SOA precursors to the TWP and can only suggest that the presence of Benzene could impact stratospheric $O_3$ via future SOA formation, which could favor novel conditions for stratospheric $O_3$ destruction according to recent hypothesis (e.g., Solomon et al, 2023).*
*In terms of VSLS, our study shows that these compounds were not transported in these pollution plumes despite their continental origin. We find that this is an important statement to keep. Other aircraft studies (different time of year, and similar area of sampling over the tropical western Pacific) showed the opposite, that VSLS (Chlorine, in particular) were indeed transported from urban regions in SE Asia to the TTL over the TWP (e.g., Treadaway et al, 2022).*

3) The back-trajectories are calculated for 40 days. This is a very long time period for air mass transport, without the consideration of diabatic processes (cloud processing), turbulence and small-scale mixing, such that the longest trajectories only represent a very crude approximation for the air mass history. Especially trajectories who travel over the Indian Ocean to Africa and from there back to the TWP region must be treated carefully. These limitations should be discussed in the manuscript.

The occurrence of convection associated with the trajectories also has to be discussed in a little more detail. As the trajectories are calculated from ERA5 dynamics, there is no guarantee that individual convective cells and elements on the order of <30 km are represented properly in the meteorological re-analysis data. Of course, the satellite data represent reality, but it should be analysed / discussed whether the convection in the re-analysis is co-located with the observed convective activity or whether this activity is sufficiently wide-spread such that air parcels in the convective region have a high probability to be lofted.

The analysis of air parcels in convection also reveals that it is not guaranteed that an air parcel actually has seen the convection, only because it has been in its vicinity: it could also happen that air parcels simply stream around the anvil of the convective towers and are not substantially modified by lofted near surface air.

*These are all very good points to consider when using trajectory analysis. In Sections 2.4 and 3.4, we point out the uncertainties associated with winds and provide references that caution against the use of single trajectories. A more robust approach to interpreting trajectory calculations is to consider trajectory ensembles, instead, which provide a more robust statistical perspective. That is the approach we felt was most suitable in our analysis. With regards to convection, our calculations are anchored to satellite observations of precipitating clouds, which require multiple considerations (see Pfister et al., 2022)*

4) The selected / presented compounds often have quite enhanced life times, especially CH4 and CO2, such that convective signals of events more than a few days ago can hardly be distinguished from background conditions. Consequently, the profiles from Fig.2 for methane and carbon dioxide provide a measure for the substantial background variability. For species with shorter lifetime, like CO, the variability is smaller, even though the regimes of convectively and non-convectively influenced air masses cannot be easily distinguished (in Fig.2). In Fig.3 some of the flights show a distinct enhanced CO in the UTLS, which could indicate convective outflow; however, after travel times of more than a week, it is unlikely, that these structures remain that clearly visible, indicating more local or regional convection and further lofting, e.g., by convectively induced gravity waves and their mixing processes.

On the other hand, the shorter lived compounds (C3H8, C2H2) hardly show enhancements in the profiles. Given their shorter chemical lifetime, the air masses must have been lofted several days prior to the observations, if only in the Northern observations any signal can be identified. This rules out transport from Africa, but indicates more regional continental pollution. Even the CO signal in the Southern observations is quite low, and hardly shows UT enhancements. This does not fit well with the analysis from Fig. 14 that the corresponding trajectories could have seen convection only two weeks ago. A separation of continental convection or biomass burning in the vicinity of convection or anthropogenic activity near the convection might elucidate the origin or missing signals a bit better. This is most obvious for the Southern observations, but would also hold true for the rest of the data.

*Our analysis focuses on data in the equatorial TTL. As mentioned in the Introduction, air masses at these locations are convective in origin, some more recent than others. Where that convection occurs is reflected in the variability observed in Fig. 2. The lifetimes of the trace gases shown in Fig. 2 ($CO2$, $CH_4$, CO, with $O_3$ added now) are indeed much longer than convective transport timescales. The magnitude of the variability observed in all tracers is primarily driven by hemispheric gradients (varying strengths of sources and sinks) as indicated by latitude (variable used for color-coding) and discussed in Section 3.1. Based on trace gas measurements of elevated pollutants (e.g., CO and various hydrocarbons), we know that those air masses originated over continents. Local convection (in our case, marine convection) would be secondary after those air masses*

*traveled from the continental source regions and entrained into the marine convective systems. This sequence of transport processes would be difficult to pin point, however.*

*In terms of strength of signal from pollution over time, it is true that the structures would evolve – likely decrease in magnitude as a result of mixing. Previous studies based on aircraft observations provided evidence that pollution plumes can indeed remain fairly distinct against the background after nearly 2 weeks of transport (e.g., Jost et al., 2004). This prior evidence makes air masses with elevated CO originating from Africa quite feasible, given the similar time scales.*

5) The MLS data does not really sheds light onto the observations, as the signal cannot be properly distinguished from the background. Therefore, the manuscript could be shortened here, and the figures 12 and 13 could be moved to an appendix or supplement.

*We think it is important to highlight the capabilities of each platform and what can and cannot be observed. Our analysis illustrates this point. For instance, if you cannot see the signal of pollution due to lower spatial resolution in your measurement (e.g., satellite measurement), that does not mean that the pollution is not present. In order to be more succinct in this section of the manuscript, we consolidated the presentation of the results to a single figure and moved additional figures to the Supplement section.*

6) The pollution is mostly determined in the cloud-free air. As the authors state, that the pollution is mostly transported for several days, and that the trajectories are slowly lofted by radiative cooling, this is not very surprising. On the other hand this is a bit contradictionary to the finding of Fig. 11, which states that the pollution is usually associated with warmer air masses. As the typical temperature profile does not increase quickly in the low latitudes above the thermal tropopause, this needs an explanation.

*Encounters of the pollution plumes were indeed mostly in cloud-free air. The statement regarding warmer temperatures relates only to air masses with 3.5 – 5 ppmv $H_2O$, which is a subset of the data, as described in Section 3.2. We revised the text to clarify this point. We also added a new panel (now Fig. 10) that illustrates warming in the wetter range of $H_2O$ and no warming in the drier range of $H_2O$.*

7) I am missing an explanation, why the pollution is not found in cloudy or moist regions. Is this because the polluted air is at higher altitude than the clouds, i.e., because of the above mentioned lofting?

*We added a new figure to the Supplement material illustrating how pollution plumes were located above the clouds, the maximum level of convective detrainment. We also included a statement in Section 3.2 highlighting a potential link between pollution and cloud formation and the need for a more comprehensive data set (e.g., physical properties and*

*chemical composition of aerosols, which are not available in our data set) to examine the radiative characteristics of these air masses.*

8) What information can be deduced from Fig. 9, which is not also included in Fig. 10? Can you (re)move Fig. 9 to the supplement, in order to slightly shorten the manuscript and reformulate the statements from the corresponding paragraph? Basically, the same conclusions, i.e., that the highest pollution levels are found in non-saturated regions, can be drawn from Fig. 10, which has better representation via the color coded visibility.
*We appreciate this suggestion and made the proposed changes.*

However, the color scale of Fig.10 (using rainbow colours) could be modified to be more perceptually uniform and color-blind / black-white friendly.
*The color scale was modified to use the "turbo" instead of the "jet" colormap.*

9) Please discuss the variability in the vertical motion of the individual trajectories: especially trajectories from Africa travel more than 50 hPa upwards and downwards on timescales of a day or two (Fig. 14c, around day 13 to 21). What processes would you associate these relatively rapid air motions including downwelling, before the radiatively driven ascent prior to the measurements?

*The rapid changes in short time scales observed in trajectories linked to Africa as a source region is a good question. That was the case for a few trajectories, but not for the bulk. As mentioned earlier, encouraged by earlier comments and suggestions, we decided to take a more statistical approach to the analysis of these calculations and revised the figure, now Fig. 12.*

References:

Jost, H.-J., Drdla, K., Stohl, A., Pfister, L., Loewenstein, M., Lopez, J. P., Hudson, P. K., Murphy, D. M., Cziczo, D. J., Fromm, M., Bui, T. P., Dean-Day, J., Gerbig, C., Mahoney, M. J., Richard, E. C., Spichtinger, N., Pittman, J. V., Weinstock, E. M., Wilson, J. C., and Xueref, I.: In-situ observations of mid-latitude forest fire plumes deep in the stratosphere, Geophys. Res. Lett., 31, doi:10.1029/2003GL019253, 2004.

Pfister, L., Ueyama, R., Jensen, E. J., and Schoeberl, M. R.: Deep convective cloud top altitudes at high temporal and spatial resolution, Earth and Space Science, 9, e2022EA002475, doi:10.1029/2022EA002475, 2022.

Solomon, S., Stone, K., Yu, P., Murphy, D. M., Kinnison, D., Ravishankara, A. R., and Wang, P.: Chlorine activation and enhanced ozone depletion induced by wildfire aerosol, Nature, 615, 260-264, doi:10.1038/s41586-022-05683-0, 2023.

Treadaway, V., Atlas, E., Schauffler, S., Navarro, M., Ueyama, R., Pfister, L., Thornberry, T., Rollins, A., Elkins, J., Moore, F., and Rosenlof, K.: Long-range transport of Asian emissions to the West Pacific tropical tropopause layer, J. Atm. Chem., 29, 81-100, doi:10.1007/s10874-022-09430-7, 2022.

Reply
**Citation**: https://doi.org/10.5194/egusphere-2024-3832-RC1

**ACPD – Referee #2 Comments**

**RC2**: 'Comment on egusphere-2024-3832', Anonymous Referee #2, 22 Jan 2025  reply

Pittman et al. present in their manuscript in-situ measurements from ATTRAX campaigns, focusing on polluted air masses in connection to biomass burning. They analyze a number of pollution trace gases with different origin or atmospheric lifetimes. Further, they use MLS CO data for contextual information for the aircraft measurements, as well as backward trajectories in order to estimate the origin of measured air masses.

The manuscript is well-written and certainly within the scope of ACP and deserves publication after addressing my remarks below.
General remarks:

1- The introduction starts with the importance to observe pollution, which may enter the upper troposphere or even the lower stratosphere because of its potential to destroy ozone, and I fully agree that such kind of measurements are very important. However, the link to ozone is in the main part of the manuscript rather weak and is only briefly mentioned again in the conclusions.
*The recovery of the stratospheric ozone layer is a central motivation for the analysis we conducted. The goal of our analysis was to elucidate what does and does not enter the global stratosphere via the tropics by exploring the chemical composition of air based on aircraft measurements. We added a new statement in Section 3.2, after the description of halogenated VSLS composition indicating the finding that the pollution plumes encountered would not have a direct effect on stratospheric $O_3$ via rapid delivery of these short-lived $O_3$ destroying compounds.*

2- Section 3 starts with the presentation of the trace gases CO2, CH4 and CO, but in the instrument description section, it was mentioned that many more trace gases have been

measured during the ATTREX campaigns. I would like to know why the authors chose to limit themselves only to these three trace gases.

*We focused on data from high frequency measurements first to best capture the unexpected airmasses that we encountered. Other trace gases were later added to further explore the chemical composition and origin of these unusual airmasses.*

3- It is not clear to me why the UCATS instrument is introduced, but the O3 measurements are never used in this work.

*Ozone measurements from UCATS are provided in Fig. 5 (rightmost panel).*

4- In general, the structure of section 3 should be better motivated (maybe in the introduction to this section?): Why are there subsections focusing on (3.1 & 3.3) CO2, CO, CH4, while others (3.2) focus on VSLS/HCs?

*This is a good suggestion. We included a paragraph at the start of Section 3 describing why and which trace gases were used within that section. While organizing this paragraph, we decided to present O3 data earlier, along with CO2, CH4, and CO (see new Figs. 2 and 3). This addition strengthens the discussion when it was based on carbon trace gases only.*

5- A number of figures in this study use the "rainbow/jet" colormap, which is not suited to quantitatively represent data. I suggest to change to a colormap, which have color changes, which are perceptually constant (e.g. "turbo", "viridis" or "inferno")

*We appreciate this suggestion. We changed all relevant figures from the "jet" colormap to the "turbo" colormap.*

Specific points:

6- Introduction: I think also studies showing stratospheric entry of polluted air masses in mid-latitudes should be briefly mentioned here. In particular about the Australian New Year fire (e.g. Khaykin et al., 2020 or Schwartz et al., 2020) and the Canadian wildfires (e.g. Pumphrey et al. 2020), which have been exceptional events.

*We appreciate this suggestion. We included these references in the Introduction.*

7- Section 2.3: The authors could mention if they have followed the recommended data screening for the MLS data (as explained in the MLS data quality document).

*We added a statement confirming proper data screening following MLS's team recommendations in Section 2.3.*

8- Line 89: Here, three remote sensing instruments are mentioned, but it seems like they are not used for this study? I suggest to briefly mention why these measurements were not used in this study.

*We adjusted the statement in 2.1 to mention the total number of instruments on the payload (not separated by remote and in situ) and a list of all parameters/variables measured during the campaign. The reader is provided with a reference for a review paper on the campaign to further learn about the payload. The remote sensing instruments during*

*ATTREX-3 were CPL, MTP, and mini-DOAS. For our analysis, we did not find their products to contribute with pertinent data.*

9- Fig. 1: I understood the text that only a subset of the ATTREX-2 and ATTREX-3 flights are used in this study, while all of the flights are shown in this figure. I suggest to differentiate between the flights used in the study and the rest of them. E.g. the colors for the flights not used could be lighter, or the lines could be dotted.
*We appreciate this suggestion. We adjusted the figure to highlight the flight tracks analyzed in our study by coloring those flights only.*

10- Line 132: Please use metric units (at least additionally to psia)
*It is common practice to use psia in instrument descriptions such as the one provided for the GWAS canisters. Per reviewer's request, we included approximate values in metric units.*

11- Line 211: "Contribution from these processes could be evaluated by examining additional trace gases such as O3": So why is O3 not analyzed here, since it is available from the UCATS instrument?
*We included new statements in Section 3 and added plots with $O_3$ data (Fig 2 and 3) to make use of the additional information provided by this trace gas.*

12- Figures 2&3: I suggest to combine these figures (and use a different colormap for latitudes and mixing ratios). I like the two perspectives on the data, but the figures are discussed together, so I would not separate them here. Further, label font sizes are very small for Fig. 3, please change it to the same style as Fig. 2.
*We made adjustments to these two figures (e.g., colormap and font sizes). We rephrased Section 3.1 by adding statements that highlight the unique information provided by the two perspectives from these figures.*

13- Figure 6: Even though it is mentioned in the caption, I suggest to label the rows accordingly directly in the figure to help the reader.
*We appreciate this suggestion and made the change.*

14- Line 268 and Figure 6: I was wondering why the gases are always listed in this order. I would have ordered it according to the atmospheric lifetime.
*This order is driven by increasing atmospheric lifetimes within three different groups: high frequency data (CO and $CH_4$), biomass burning trace gases ($C_3H_8$, $C_2H_2$, and $CH_3Cl$), and industrial sources ($C_2Cl_4$). We added a few descriptive words at the start of the second paragraph in Section 3.2 to justify their use in the analysis.*

15- Line 303: "In all instances, HCs and CH3Cl were positively correlated …": I think that one can only guess the correlation based on the plots provided, and only for the NH bin (where a better suited plot is provided in Fig. 7, which is not mentioned here). For the other bins, data points are sparse (in particular the pollution points), so I find it hard to see a

correlation there.

*We revised the text to clarify that Fig. 6 shows qualitative correlations. The next step was to quantify the correlations, which is the focus of Fig. 7.*

16- Figure 7: The grey open circles are hard to see at all, they are much too small. A legend explaining the different shapes of the points would also help (even though it is mentioned in the caption). Did I understand this correctly, that Fig. 7 only shows a subset of the data used in Fig. 6? Further, it would help to remind the reader on the approximate lifetimes of the substances shown in the panels, since they are discussed in the text.

*We revised Fig. 7 to include the suggestions provided here.*
*The data used in Fig. 7 is the same as Fig. 6, but capped at 17 km to exclude the stratosphere in the enhancement ratio calculation.*

17- Figure 9: I'm not sure if I have understood the benefit of this figure. What arguments are based only on this figure, which cannot be explained by Figure 10? I think Figure 9 could be moved to a paper supplement.

*We retained the information provided by both figures in the text and opted for keeping only Fig. 10.*

18- Line 383: "No correlation between temperature and CO was observed at these low H2O levels.": I think there is a weak (anti-)correlation with temperature in the plot, at least the points are getting darker blue for warmer temperatures?

*This is a correct observation that holds true in the absence of pollution (elevated CO). The pollution plumes we encountered showed a significant increase in CO while temperatures remained fairly stable (~191 K) in the low $H_2O$ regime. We opted for removing the statement brought up in this comment, in order to avoid confusion. We added a new statement and a sub-panel with histograms of temperature in this low $H_2O$ regime in (new numbering) Fig. 10 to better illustrate the relation between temperature and high CO.*

19- Line 384: "The second regime is air masses with H2O above 5 ppmv ...": Why is the threshold chosen to be 5 ppmv and not (e.g.) 4.5 ppmv?

*We included a statement that addresses this concern. The choice of thresholds was visually determined from the figure. The values are simply used to guide the discussion of distinct regimes observed in the data.*

20- Line 392: "These warmer and polluted air masses were encountered 15 % of the time in the 3.5 to 5 ppmv H2O range and at altitudes between 16 and 17 km.": I do not understand this sentence, please rephrase.

*We rephrased the statement. The goal of the statement was to point out that those particular air masses were not encountered very often.*

21- Line 513: "A ten-year MLS record showed...": Maybe I missed it, but there was no 10-year MLS record shown or mentioned in this manuscript yet? I think this thought should be discussed before the conclusions section.

*The statement of the 10-year MLS record first appears at the end of the first paragraph in Section 3.3. The reader is directed to a figure showing that data in the Supplemental section (Fig S4).*

22- Line 550: Please also mention where to find ERA5 data (used for the trajectories). Further, is the trajectory model (a specific name is not mentioned in the manuscript) used in this work available? Do you have a website for that?
*We included the link where ERA Interim fields can be obtained.*
*Description and references for the trajectory model were provided in Section 2.4. The model itself is not publicly accessible.*

Reply

**Citation**: https://doi.org/10.5194/egusphere-2024-3832-RC2